

# Broadcasting climate change: An international survey on weather communicators' approaches

Tomas Molina [1]* and Ernest Abadal [2]

[1] Applied Physics, Universitat de Barcelona, Spain
[2] Universitat de Barcelona, Spain

*Correspondence to*: Tomas Molina (tomasmolinabosch@ub.edu)

**Abstract.** This study explores the role of television meteorologists as key communicators of climate change across diverse global contexts. Utilizing a survey of 204 participants from 81 countries, it examines their perspectives, strategies, and

challenges in addressing climate issues. Respondents, predominantly experienced professionals with meteorological and media expertise, highlighted the communicative potential of television weather segments, despite their brevity. Most participants reported strong climate knowledge, significant concern about its impacts, and reliance on trusted scientific sources like IPCC reports. Key barriers to effective communication included the complexity of climate science, misinformation, and limited public understanding. However, respondents identified strategies for improvement, such as tailored messaging, engaging

visuals, and leveraging social media to reach younger audiences. Television emerged as the most impactful medium for raising awareness, complemented by print and digital platforms. The findings underscore the need for a global communication strategy emphasizing clear, actionable, and solutions-oriented messaging. By aligning international efforts with localized approaches, television meteorologists can play a pivotal role in bridging scientific insights and public engagement. This research highlights the critical importance of fostering informed societies, enacting national regulations, and advancing international agreements

to drive collective action against climate change.

## 1 Introduction

Climate change is a pressing global challenge with profound implications for all species on Earth, driven in large part by human activities. Addressing this crisis requires a multifaceted approach, with effective communication emerging as a critical tool for fostering awareness and inspiring action. Among these efforts, improving the strategies for communicating climate

change on an international scale remains a top priority. The media, as a dominant source of information in many countries, has naturally positioned weather segments as key platforms for discussing climate-related issues. Television meteorologists, with their established roles as skilled communicators, have long been effective in delivering weather and climate information to broad audiences. Often regarded as "station science communicators" (Hochachka, 2022; Wilson, 2008), this study seeks to explore their perceptions and evaluations of climate change communication.

The challenge of effectively communicating climate change encompasses a wide range of stakeholders, including social, political, and economic actors (Castree et al., 2014; Osaka et al., 2021). As globalization intensifies, the need for a coherent





and internationally aligned communication strategy becomes increasingly apparent (Samuel Craig & Douglas, 2001; Voola et al., 2022). Communicators face significant hurdles, given the complexity of the subject matter, which includes technical and scientific data, politically sensitive issues, and information sourced from a diverse array of entities, such as government

agencies, academic institutions, blogs, and the ever-expanding landscape of social media (Iyengar & Douglas S. Massey, 2019; Schäfer, 2012).

The media often mediates public perception of climate change through news programs, debates, and coverage across various platforms (Bora Abhijit, 2012; David, 2022). However, mainstream media content is frequently shaped by the need to simplify, con-dense, and entertain, which can undermine the nuanced communication of scientific issues (Hase et al., 2021; Perloff,

2009). Studies indicate a trade-off between scientific depth and audience appeal, with higher media ratings often correlated with less scientific content (COSCE, 2005; Riffe et al., 2023). Television producers also tend to allocate scientific topics to specialized channels, limiting their reach to niche audiences. As a result, communicating the complexities of climate change including mitigation, adaptation, and reduction strategies poses distinct challenges (Nerlich et al., 2010; Verlie, 2021). These difficulties are exacerbated by the probabilistic and statistical nature of climate science and the varied degrees of expert

consensus, which complicate messaging strategies.

Notably, most media coverage of climate change is integrated into general news services rather than specialized scientific reporting. Newsrooms, often staffed with experts in politics, economics, and culture, seldom prioritize scientific journalism, leaving coverage of climate issues to general reporters. Such coverage typically emerges in response to specific news events and is frequently incorporated into weather segments, where the breadth of the issue may not be fully addressed (Howarth &

Anderson, 2019). Despite their brevity ranging from 50 seconds to two and a half minutes (Molina, 2005) television weather segments hold significant communicative potential due to their frequent airing and wide viewership. This positions meteorologists, who often possess scientific training, as trusted voices on climate-related issues and de facto "resident scientists" within their networks (Henson, 2013; Rainear & Lachlan, 2022).

While numerous studies have examined the role of television meteorologists in communicating climate change, these have

largely focused on national contexts, particularly within the United States (ACOMET, 2017; E. W. Maibach et al., 2020). In the U.S., the field benefits from strong professional and scientific networks (Farnsworth & Lichter, 2012; A. A. Leiserowitz et al., 2013; E. Maibach et al., 2017; Schäfer, 2012). However, there is a paucity of research that explores the international dimensions of this issue, especially considering the diverse socio-economic, cultural, and developmental contexts in which meteorologists operate globally.

This study seeks to address this gap by examining how television weather reporters worldwide perceive and approach climate change communication. It aims to identify universally effective communicative strategies, including messages, indicators, channels, and the role of meteorologists as communicators. The research will also investigate the sources of information used, the evolution of media handling of climate issues, and emerging trends. By adopting a global perspective, this study underscores the importance of an international communication strategy that complements and enhances national efforts to

address climate change effectively (Molina, 2025).





## 2 Materials and Methods

This study employs a qualitative research approach, using a survey to examine the perspectives of television meteorologists on climate change communication. Due to the absence of a universal census for this professional group, we constructed a non-representative sample based on contacts from three major international organizations:

1. International Association of Broadcast Meteorology (IABM): A global network of television meteorologists affiliated with the World Meteorological Organization (WMO) and the Intergovernmental Panel on Climate Change (IPCC).
2. International Weather Forum (FIM): An annual event promoting meteorological education and outreach.
3. Climate Without Borders (CWB): A network facilitating real-time information exchange among weather presenters.

From these organizations, we compiled 952 contacts: 615 from IABM, 182 from FIM, and 155 from CWB. The survey was administered in English to address linguistic diversity and included demographic, knowledge-based, and opinion-oriented questions about climate change communication. It utilized Likert scales and open-ended questions to explore nuanced views, with neutral responses analysed separately to focus on clear trends.

The questionnaire was piloted with international reviewers to ensure clarity and reliability, particularly on sensitive topics. It was distributed online via Google Forms, ensuring accessibility across various devices and operating systems. Initial distribution occurred via WhatsApp and email, supplemented by targeted sharing within professional networks in countries such as the U.S., Canada, Croatia, and Spain.

While the survey does not aim for statistical significance, its objective is to gather diverse insights from television meteorologists worldwide, ensuring broad geographic and cultural representation. The questionnaire and its responses are available in the supplementary materials.

## 3 Results

The survey received 204 responses from 952 contacts (21.4% response rate) across 81 countries, highlighting global distribution, although participation was lower in North-Central Africa and the Middle East (Figure 1). Most respondents were daily con-tributors to their national television stations, aligning with their significant influence on widely viewed news programs (Fleming, 2005). The survey participants' responses regarding the best resources, channels, messages, and indicators for communicating climate change to broader audiences are presented in Table 1.





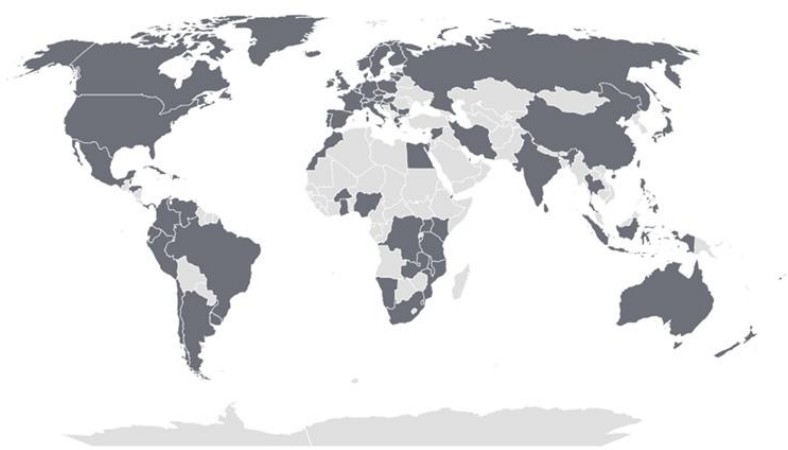

**Figure 1: Country distribution of the replays to the survey.**

### 3.1 Participant Demographics

The survey respondents were 61.1% male and 38.4% female, with one participant opting not to disclose their gender. Most were over 46 years old (55.1%), with extensive experience—78.8% had over 10 years in the field, and 47.3% had over two decades (Figure 2). Half identified television as their full-time occupation, while others worked intermittently in media. Many engaged on social media platforms, including Twitter (69.5%), Facebook (68%), and Instagram (50.2%), alongside professional tools like WhatsApp groups (47.3%).

Meteorology was the most common educational background (71.9%), followed by journalism (18.2%), with all participants holding university-level qualifications.

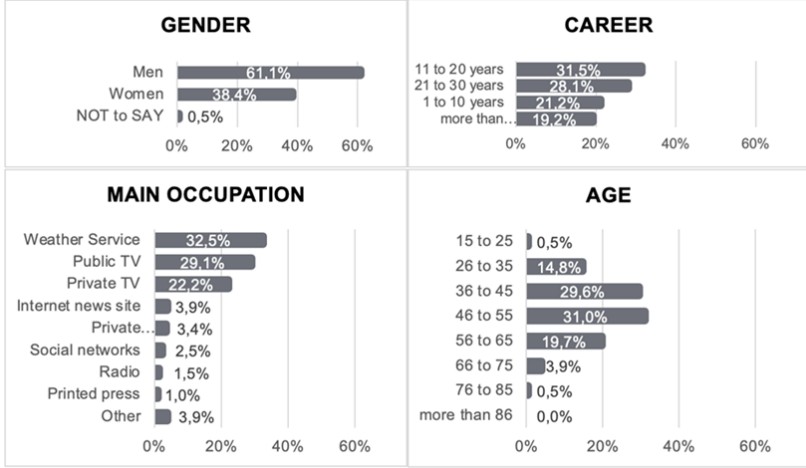

**Figure 2: Gender, career, main occupation, and age of survey participants.**



## 3.2 Knowledge and Perception of climate change

Most participants rated their climate change knowledge as good or very good (85.2%), with slightly lower confidence in regional knowledge (82.2%) and reporting expertise (73.9%). While 86.2% dismissed climate change as a hoax, 13.8% expressed some un-certainty. A majority (89.2%) observed recent climate changes in their areas, predominantly with negative
impacts (65.5%).

On personal feelings about climate change following the Six Americas scheme of Yale and George Mason Universities (A. Leiserowitz, 2011), 68% expressed concern, 23.6% felt alarmed, and 7.4% were cautiously concerned. Commonly cited indicators of climate change included temperature changes, precipitation shifts, and extreme weather events, with societal impacts such as agricultural and wildlife changes (Figure 3).

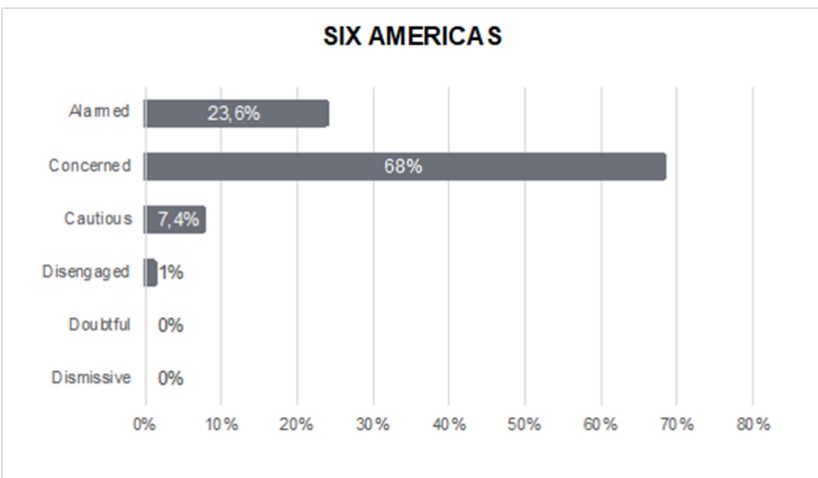


**Figure 3: Survey participants feelings about climate change.**

## 3.3 IPCC Reports and Information Sources

About 89.4% of respondents had read at least one IPCC report, with the Special Report on Global Warming of 1.5°C being the most widely read (74.9%). The majority found IPCC reports reliable (76.3%) and scientifically sound (78%) but noted their
limited societal in-fluence and perceived political bias (51.7%).

Communicators preferred official reports (66.5%) and scientific journals (65.5%) for in-formation, with television and social media being less favoured.

## 3.4 Climate change in Media

While 72.9% of communicators reported on climate change, only 37.9% believed their audiences were well-informed. The
main barriers included the complexity of the topic (61%) and misinformation campaigns (51.2%). Coverage was typically event-driven, focusing on extreme weather or new studies, and challenges included a lack of media emphasis on the issue's urgency.




### 3.5 Strategies for Effective Communication

Effective communication strategies included using tailored language, clear visuals, and engaging younger audiences via social
media. Communicators emphasized realistic messaging (88.1%) and collective responsibility, identifying governments, industries, and society as key actors. Television (91.6%) was deemed the most effective awareness channel, followed by traditional and digital media.

### 3.6 Facilitating climate Action

A well-informed society (97%) was seen as vital for progress, along with national regulations (92.1%) and international
agreements (91.1%). Communicators highlighted the need for decisive government action, public engagement, and shifts in energy resources to address climate change effectively.

| RESOURCES | | CHANNEL | |
|---|---|---|---|
| A more informed society | 97 % | Television | 91.6% |
| National regulations | 92.1% | General press | 88.1% |
| International agreements | 91.1% | Websites in Internet | 83.7% |
| Technology | 87.1% | New content in new media | 80.7% |
| Changes in the economy | 87.1% | Social Networks | 80.2% |
| Penalties and fines | 64% | Every citizen speech | 63.5% |
| Progress is already being made correctly | 19.1% | Plays and movies | 62.5% |
| | | Artistic works | 62% |
| | | The religious leaders' sermon | 39.9% |

| MESSAGE | | BEST INDICATORS | |
|---|---|---|---|
| A realistic message | 88.1% | Changes in sea level | 93.5% |
| We can reduce Climate Change | 87.1% | Ambient temperature | 92.6% |
| A change in the economic model is necessary | 85.2% | The amount of water available to society | 80.2% |
| Technology alone will not solve the problem | 77.3% | Invasive species | 76.3% |
| A message of hope | 74.3% | The atmospheric pollution | 63.5% |
| We are in a Climate emergency | 76.8% | The quality of water available to society | 55.1% |
| We are in a Climate crisis | 75.8% | Pollution of rivers and sea | 54.6% |
| We are in a climate catastrophe | 34.4% | The degree of social alarm | 44.8% |
| A message that scares people | 16.7% | | |

**Table 1: Survey participants opinions on best resources, channels, messages & indicators.**



## 4 Discussion and Conclusions

This survey, while not statistically representative, offers valuable insights from television meteorologists across over 40% of the world's nations, many working with major national broadcasters, most with long professional on air careers, in populous countries. The data highlights the diversity and global relevance of perspectives on climate change communication among TV meteorologists. Comparatively, similar surveys have captured smaller, localized samples, emphasizing the broad engagement achieved here (ACOMET, 2017; Hayes, 2015; A. Leiserowitz et al., 2021).

### 4.1 Effective Strategies for climate change Communication

Respondents identified three key resources for addressing climate change, from an international point of view: fostering an informed society, national regulations, and inter-national agreements. Technology and economic reforms also ranked highly. Television remains the most effective channel for reaching the public, followed by print media, the internet, and social networks. Communicators emphasized realistic, solution-focused messaging and advocated for shifting economic models to enable climate. Critical indicators of climate change include rising sea levels, air temperatures, and water availability.

Open-ended responses highlighted the importance of audience-specific language, clear visuals, and education-focused communication tailored to local realities. Coordinated strategies, as emphasized in similar studies across various media platforms, underscore the critical role of communicators' knowledge and expertise in fostering a more impactful social dialogue on climate change (León et al., 2023).

### 4.2 Perceptions and Concerns Among Communicators

Most respondents expressed concern (68%) or alarm (23.6%) about climate change, with levels of concern significantly exceeding national averages, particularly in the U.S. This heightened awareness may stem from their professional roles and affiliations with climate-focused organizations (A. Leiserowitz et al., 2021). Despite varying beliefs, all respondents acknowledged observing climate change impacts in their work, underscoring a consensus on its professional relevance. Recent literature reviews emphasize the critical role of affect and emotions as key drivers of climate change perception and action (Brosch, 2021).

### 4.3 Rising Awareness and Media Evolution

Awareness of climate change among TV meteorologists has grown since the IPCC's first report in 1990. Initially, only one in four communicators engaged with climate findings; now, three in four actively follow IPCC reports. The media has evolved from framing climate change as an abstract concern to addressing it as a crisis, supported by initiatives like Covering Climate Now and the UN's Climate Neutral Now. This shift reflects the in-creasing urgency and visibility of climate issues globally. Fostering public engagement is essential for addressing climate change, with climate communication serving as a pivotal tool in this effort (Kumpu, 2022).



**4.4 Communication Strategies for Global Awareness**

Developing international communication strategies that transcend cultural and national boundaries is vital for fostering a unified global response to climate change. Effective communication should focus on translating scientific findings into accessible, actionable messages tied to tangible impacts like extreme weather and daily life climates. Recent literature reviews highlight the vital role of communicators and scientists in bridging cli-mate knowledge to the public, fostering awareness and

understanding (Busch Nicolaisen, 2022). Television, print media, and the internet remain the most effective platforms for public engagement, despite challenges posed by misinformation and social media polarization.

Addressing climate change requires collaboration across society, governments, and the private sector, as highlighted by the UN Climate Change Conferences. Television communicators play a critical role in bridging gaps between scientific findings and public understanding, fostering a more engaged and proactive society. This aligns with recent studies on the state of climate

change communication from a global perspective, particularly emphasizing the context and challenges faced by the Global South (Schäfer & Painter, 2021).

Our study underscores the importance of leveraging television and other media to communicate climate effectively. By focusing on actionable, localized, and solutions-oriented messaging, communicators can drive greater public awareness and engagement. Coordinated global efforts are essential to address the challenges posed by climate change, making collaboration

among all stakeholders' imperative for meaningful progress.

**Ethical statement**

All participants in this research were fully informed about the study's purpose, procedures, and potential implications. They provided their voluntary consent to participate in accordance with ethical guidelines.


**Author contributions**

Conceptualization, T.M. and E.A.; methodology, T.M. and E.A.; validation T.M. and E.A.; formal analysis, T.M. and E.A.; data curation, T.M.; writing, T.M.; review and editing, T.M. and E.A. All authors have read and agreed to the published version of the manuscript.

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
