# Peer review of "Broadcasting climate change: An international survey on weather communicators' approaches"

_EGUsphere, 2025_

## Author Response (AR2)

Dear Editor, many thanks for your feedback. As you can see at the "Author's response" of 28th july this was our explanation to the question of the editor:

Thank you for your thoughtful responses to the reviewer comments and for the revisions made so far. I appreciate the importance of your topic and the potential contribution of this work and want to make sure that the methods are described with sufficient clarity in the final version. In particular, the manuscript refers to the study as a "survey" while also characterizing it as "qualitative." While qualitative surveys are possible, they are relatively uncommon and require careful explanation. Most of the results are presented in quantitative terms (e.g., percentages), which suggests a quantitative design. Importantly, using a non-random sample does not make a study qualitative—it remains a survey with a non-random sample. The manuscript needs a clearer and more accurate account of the research design, data collection, and analysis methods so that readers can properly evaluate the study.

**Thank you very much for your kind comments.**

Our study employs a quantitative approach, using a structured survey with closed-ended questions distributed to meteorologists worldwide. In the absence of a global census or sampling frame for this professional group, we adopted a non-probability, self-selected sampling method. Consequently, the sample is not statistically representative of the global meteorological community. However, the respondents' clear interest in the topic offers valuable insights and enables the identification of emerging patterns, perceptions, and trends among professionals actively engaged with the subject.

As requested by you and one of the reviewers, we have clarified this in lines 72–74.

. . .

In addition, and in response to your latest comments:

To strengthen this section, please address the following:

- Clarify whether the study is intended to be quantitative, qualitative, or mixed methods and the methods followed. For example, if it is qualitative, explain how qualitative data were collected, analyzed, and interpreted; whereas, if it is quantitative, describe how the survey instrument was developed and analyzed. The methods as they are stated now do not have enough detail for the reader to best contextualize the results, especially in a format typical of this type of research.

Our paper is based on quantitative research.

- Explain why the study is described as "qualitative" when most results are presented as quantitative measures (e.g., percentages) - in other words, can you share a little more about the methods and approach to analysis.

There is no mention of qualitative research in our paper. Our study is entirely based on quantitative research.

- Clarify that using a non-random sample does not make a study qualitative - if applicable, describe how the non-random sample was selected and any implications for interpretation.

Our paper is based on quantitative research. We have clarified this in the manuscript with the following statement:

"This study employs a quantitative research approach, using a survey to examine the perspectives of television meteorologists on climate change communication. Due to the absence of a universal census for this professional group, we constructed a nonrepresentative sample based on contacts from three major international organizations (Moniruzzaman Sarker & AL-Muaalemi, 2022)."